# Design of a Low-Power Delay-Locked Loop-Based 8× Frequency Multiplier in 22 nm FDSOI

**Naveed \* and Jeff Dix** 

Department of Electrical Engineering and Computer Science, University of Arkansas, Fayetteville, AR 72701, USA
\* Correspondence: naveed@uark.edu

**Abstract:** A low-power delay-locked loop (DLL)-based frequency multiplier is presented. The multiplier is designed in 22 nm FDSOI and achieves 8× multiplication. The proposed DLL uses a new simple duty cycle correction circuit and is XOR logic-based for frequency multiplication. Current starved delay cells are used to make the circuit power efficient. The circuit uses three 2× stages instead of an edge combiner to achieve 8× multiplication, thus requiring far less power and chip area as compared to conventional phase-locked loop (PLL) circuits. The proposed 8× multiplier occupies an active area of 0.09 mm$^2$. The measurement result shows ultra-low power consumption of 130 μW at 0.8 V supply. The post-layout simulation shows a timing jitter of 24 ps (pk-pk) at 2.44 GHz.

**Keywords:** WSN; frequency multiplier; XOR; FDSOI 22 nm

## 1. Introduction

The development of wireless sensor networks (WSNs) has seen an increased demand in the last decade. The interest can be attributed to their cost-effective and easy implementation in a wide range of fields such as agriculture, environment monitoring, surveillance, etc. [1]. Designing a sensor node requires several critical design considerations such as form factor, network size, operating conditions, power consumption, maintenance, etc. For WSNs, when designing a battery-less sensor node, minimizing the power consumption is a challenging task. Among the several functional blocks of a sensor node, most of the available power is used for carrier signal generation for data transmission. Using a local oscillator for carrier generation not only necessitates a significant amount of power consumption, but it is also quite difficult to achieve sufficient accuracy over process–voltage–temperature (PVT) variations [2]. Thus, it is challenging to implement a low-power wireless communication architecture in low-cost WSNs without the availability of a stable reference frequency.

Backscattering the incoming signal to eliminate the need for carrier signal generation has been a popular and uncomplicated solution [3]. However, backscattering can make the system prone to self-jamming [4]. The phase-locked loop (PLL) architecture is another dominant choice for carrier frequency synthesis. Ref. [5] shows the implementation of a transceiver that uses the received 915 MHz signal as input of a PLL to realize a 2.4-GHz RF carrier for wireless data transmission. However, due to the need for a VCO, a phase detector, and a frequency divider, a PLL is physically large and consumes a significant amount of power [6]. The PLL also suffers from phase noise accumulation in the voltage-controlled oscillator (VCO) [7].

In the last decade, the delay-locked loop (DLL)-based frequency synthesis has been under exploration [8–13] due to its low power, low complexity, and area-efficient performance. Ref. [8] presents a DLL based on a voltage-controlled delay line (VCDL) and an edge combiner. This approach may suffer from duty cycle distortion due to the possible mismatches in the VCDL and because the frequency multiplier triggers on both the rising and the falling edge. Another DLL uses VCDL, which can be configured as a resettable VCO [9].

However, this approach may have a high in-lock error due to the process of injecting back the frequency into the VCDL. Since the DLL operation does not require any inductors and for the most part consists of digital logic circuits, its implementation is area efficient. This paper presents the design of a low-power XOR logic-based DLL. The proposed DLL is designed as a part of a battery-less wireless sensor node. To conserve power, the sensor node extracts the carrier of an ISM band signal and divides the received frequency. The DLL then applies 8× multiplication to the input signal to generate a 2.44 GHz signal in the output. Therefore, it is critical for the DLL to be energy efficient for its application. The design uses current-starved voltage-controlled delay cells (VCDLs) to minimize power consumption. A series of three 2× multiplication stages are used to achieve 8× frequency multiplication. A new charge pump (CP)-based duty cycle control loop (DCCL) is implemented in each multiplication stage to achieve low duty cycle distortion over PVT variation.

This paper is organized as follows: Section 2 describes the implemented DLL architecture. The circuit implementation is discussed in Section 3. Section 4 presents the experimental results, and a conclusion ends the paper.

## 2. DLL Architecture and Operation

The block diagram of the proposed frequency multiplier is shown in Figure 1.

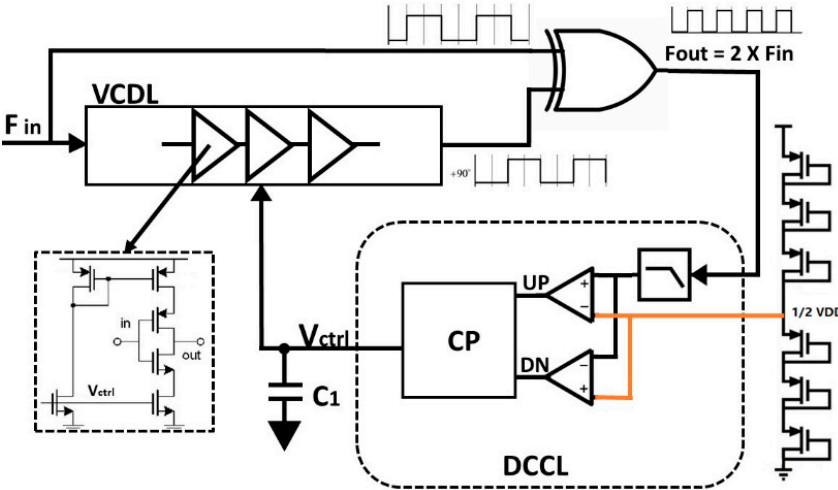

**Figure 1.** Proposed DLL architecture showing the full operation of 2× multiplication.

Each 2× multiplication comprises a VCDL, an XOR gate, and a duty cycle control loop (DCCL). A series of three 2× multipliers is used to achieve 8× frequency multiplication. VCDL is then followed by a low-power XOR gate. Given that frequency locking is not an issue, an XOR gate is preferred as the phase detector in our approach. The XOR gate takes in both the original signal and the delayed signal to output a 2× frequency, as shown in Figure 2.

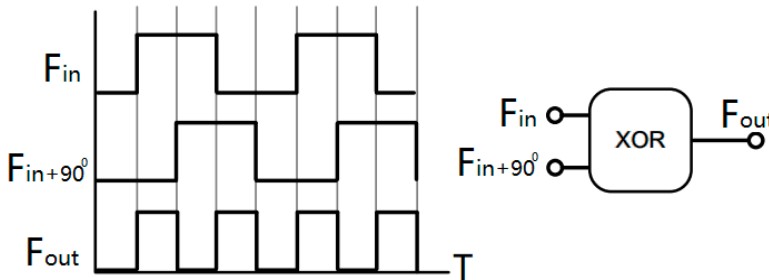

**Figure 2.** XOR logic-based frequency multiplication technique.

Initially, the delay introduced by the VCDL may not be exactly 90°. The duty cycle may be less than or greater than 50%. The duty cycle correction circuit uses a passive integrator to generate an average DC voltage proportional to the duty cycle. The output of the integrator is compared with a voltage level equal to half of the supply voltage (VDD) using comparators. After comparison, the DCCL circuit signals the charge pump (CP) circuit to generate feedback control voltages. The feedback coming from CP then varies Vctrl to correct the delay to 90°, achieving a 50% duty cycle. It is to be noted that in the case of a single 2× multiplication stage for an input frequency of 305 MHz, the whole circuit except the output of the XOR gate is switching at 610 MHz, i.e., twice the input frequency.

## 3. Circuit Implementation

### 3.1. Current-Starved Delay Cell

Current-starved full-swing inverter cells are used in the VCDL to generate the 90° delay. The implemented delay cell is optimized for low power consumption and is shown in Figure 3. The current in the delay cell is controlled by the gate voltage of MN2 and MP2. The inverter is sized to produce only a fraction of the total delay. Too many or too few delay cells in VCDL can impact the process corner variation. Therefore, the number of delay cells in each VCDL is optimized to achieve better performance across process corners. Careful consideration in layout design is taken to minimize mismatches.

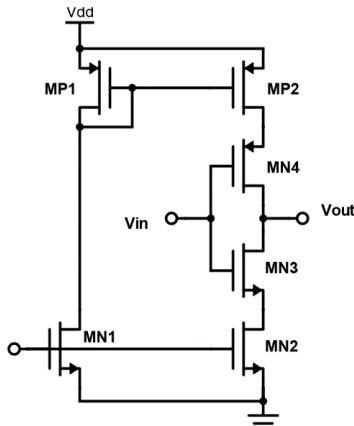

**Figure 3.** Current-starved delay cell.

### 3.2. XOR Logic

Conventional XOR gates are used for frequency multiplication. The use of an edge combiner is avoided in this implementation to reduce the number of XOR gates. The current implementation uses only three XOR gates in total to achieve 8× frequency multiplication. Comparatively, an edge combiner will require seven XOR gates for 8× multiplication. The schematic of the implemented XOR gate topology is shown in Figure 4. Since each multiplication stage has a different input frequency, each of the XOR gates is optimized for power consumption and frequency.

### 3.3. Duty Cycle Correction Loop

The DCCL ensures a 50% duty cycle of the output on all PVT corners. It controls the delay of VCDL using the signal Vctrl. The feedback forces the average (i.e., DC component) of the 2× signal to be equal to half of the VDD to achieve a 50% duty cycle. To save power and area, diode-connected stacked PMOS devices are used. The stacked MOS diodes generate a VREF equal to half of VDD. A passive RC integrator extracts the DC of the 2× signal and compares it with VREF using OTAs, as shown in Figure 5. The OTAs are biased in the subthreshold region to achieve ultra-low-power operation. The UP and DN signals coming from the OTAs are fed into the charge pump circuit to generate Vctrl.

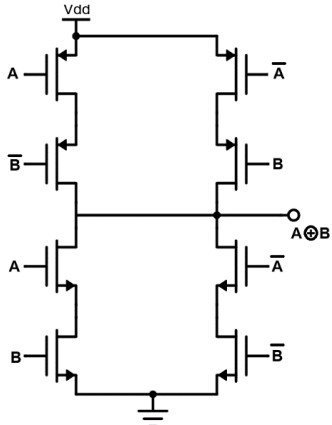

**Figure 4.** Implemented XOR gate.

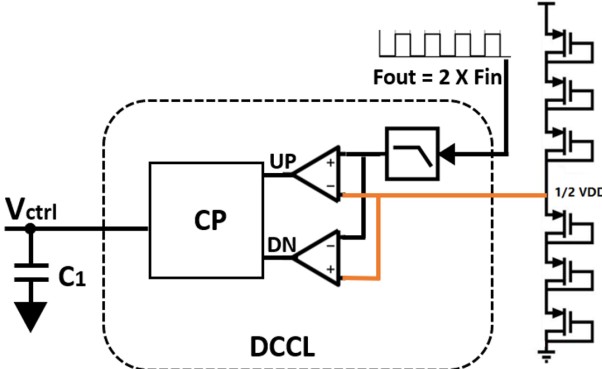

**Figure 5.** Proposed duty cycle correction loop.

The implemented charge pump (CP) circuit is shown in Figure 6. The CP is biased in the subthreshold region for low current consumption. Self-cascoded transistors are used in the current mirrors to boost the output resistance allowing a high output voltage swing. When the duty cycle is >50%, the UP signal is High, and the DN signal is Low so that C1 charges and Vctrl increases. This increase in Vctrl decreases the delay in the VCDL until the duty cycle becomes 50%. When the duty cycle is <50%, the UP signal is Low and the DN signal is High, thus correcting the duty cycle by decreasing Vctrl. This feedback loop ensures a 50% duty cycle across all PVT corners.

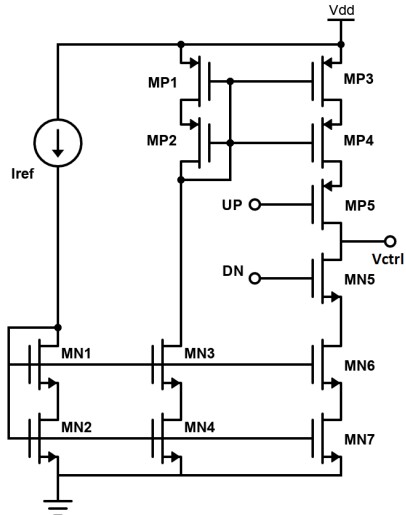

**Figure 6.** Charge pump circuit implementation.

### 3.4. Simulation Results of a Single 2× Multiplication Stage

Figure 7 illustrates a transient simulation of a single stage of 2× frequency multiplication. The input frequency provided to the 2× multiplier is 305 MHz, yielding a 605 MHz signal at the output. Due to the action of the duty cycle correction loop, the circuit achieves a 50% duty cycle across all simulated process corners. The simulated power consumption of the 2× multiplier is approximately 40 µW at a 0.8 V supply.

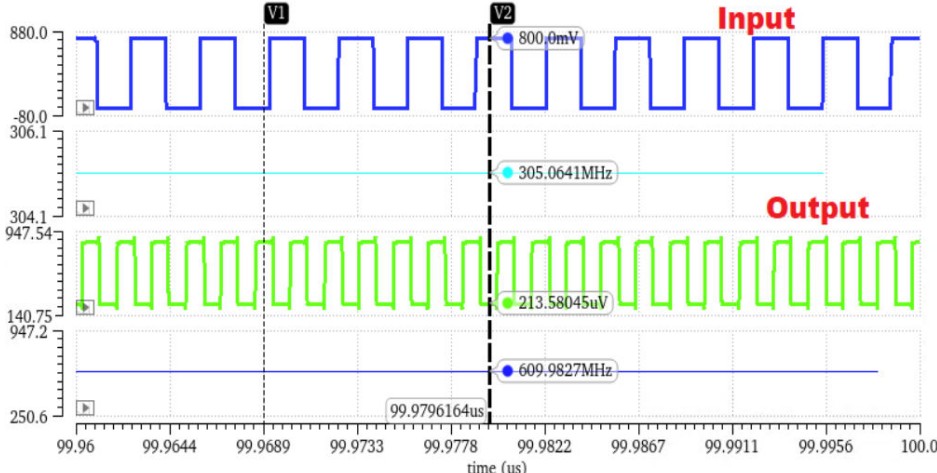

**Figure 7.** Simulation results of the input vs. output frequency of the 2× multiplication stage.

For the implementation of DLL in this work, 22 nm FD-SOI (Fully Depleted Silicon-On-Insulator) process technology is used. FD-SOI transistor technology offers superior electrical performance due to its reduced parasitic capacitance, energy efficiency, and improved isolation as compared to the transistor fabricated over bulk silicon [14], leading to enhanced signal integrity and lower power consumption. Additionally, the improved short-channel effects in FD-SOI contribute to better device scaling, allowing for the integration of more compact and efficient DLL components. Overall, the utilization of the FD-SOI process in DLL implementation promises superior performance, reduced power consumption, and increased design flexibility compared to traditional CMOS processes.

### 4. Experimental Results

The circuit is implemented in 22 nm FD-SOI technology and is designed in Cadence Virtuoso. The block schematic of the 2× multiplier is shown in Figure 8 which highlights all the sub-blocks of the circuit. The proposed 8× multiplication circuit occupies an active area of 0.09 mm². The layout and micrograph of the circuit is shown in Figure 9.

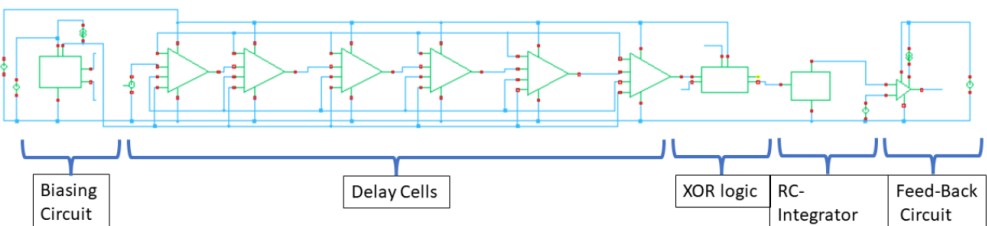

**Figure 8.** Block schematic of the 2× multiplier in 22 nm FD-SOI.

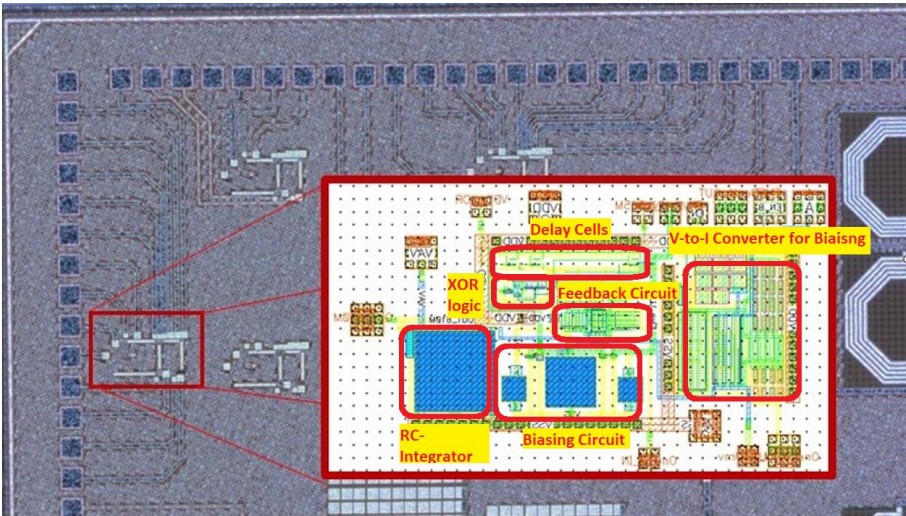

**Figure 9.** Micrograph and layout of the proposed circuit.

Figure 10 shows the post-layout performance of the 8× frequency multiplier. Here, 305 MHz is given as input frequency to the multiplier, which generates an output frequency of 2.44 GHz. Figure 11 shows the action of the control loop to correct the duty cycle of the output signal. It takes about 40 µs to achieve a 50% duty cycle. The measured power consumption of the 8× multiplier is about 130 µW at 0.8 V.

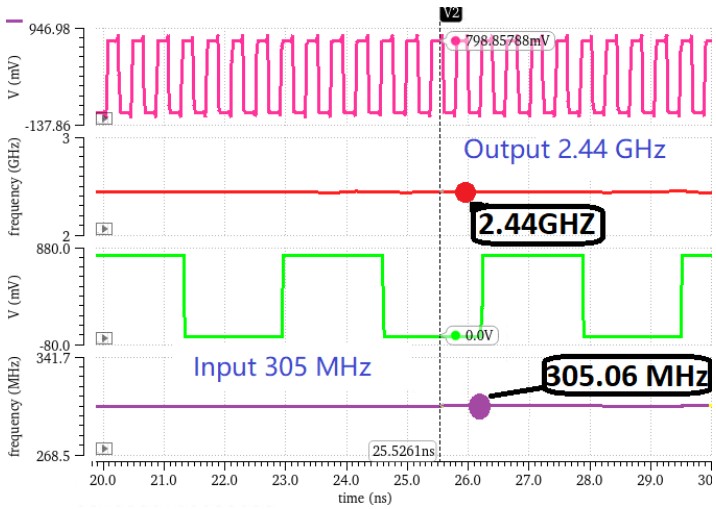

**Figure 10.** Post−layout simulation of the input vs. output frequency of the proposed circuit showing 8× multiplication.

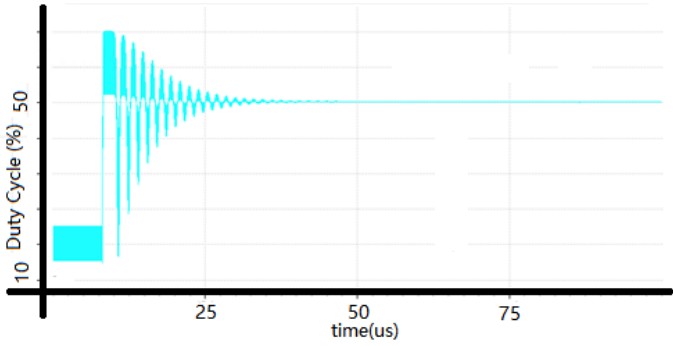

**Figure 11.** Post layout simulated response of the duty cycle correction loop. The circuit takes 40 µs to achieve a 50% duty cycle.

The circuit passed all the post-layout process corner simulations for a temperature range of −20 to 110 °C while the supply voltage was kept constant at 0.8 V.

The DC offset of the OTAs for error corrections changes across process corners and appears as a static duty cycle error. This error can be seen as spurious tones in the output frequency spectrum. For all process and temperature corners, the duty cycle error remains in an acceptable range of −5% to 3%. For all the process and temperature corners, the number of delay stages is chosen such that the duty cycle error before correction is always positive. The observation shows that the feedback loop can correct a max duty cycle error of 30% before saturating.

Figure 12 shows the measured frequency spectrum of the output. The sideband harmonics power level is seen at least 25 dB lower than the carrier. The simulation shows a timing jitter of 24 ps (pk-pk).

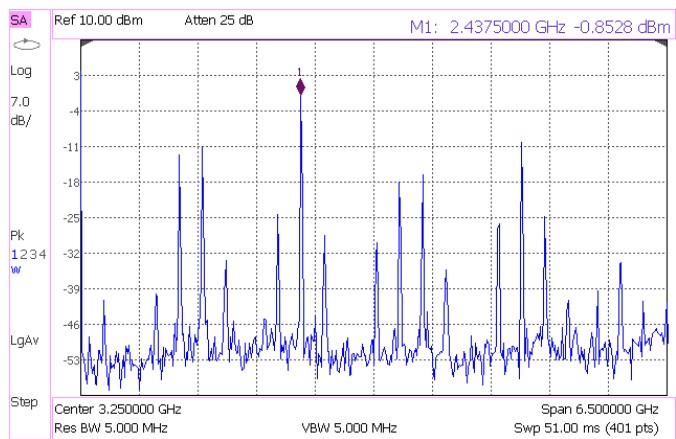

**Figure 12.** Measured frequency spectrum of the output.

A FOM (figure of merit) is derived from [15] to make a performance comparison with the state of the art. Equation (1) calculates the FOM using multiplication factor (N), Power Consumption (Pdc), Process minimum length ($L_{min}$), operating bandwidth (BW in %), and Area (A). The comparison of performance is summarized in Table 1. The proposed multiplier achieves the best FOM due to lower power consumption, wider operating bandwidth, and better multiplication factor/area ratio. Note that a lower FOM is an indication of better performance.

$$FOM = 10 \log \frac{Pdc}{N} + 10 \log \frac{A}{L_{min}} - 10 \log(BW), \tag{1}$$

**Table 1.** Summary of performance comparison.

|  | **This Work** | **[11]** | **[12]** | **[13]** |
|---|---|---|---|---|
| Supply Voltage (V) | 0.8 | 0.6–1.2 | 1 | N/A |
| Multiplication Factor | 8× | 32× | 3× | 3× |
| Input/Output Frequency (GHz) | 0.305/2.44 | 0.017/0.574 | 20/60 | 3.5/10.5 |
| Timing jitter (Simulated) (ps) | 24 ps @ 2.44 GHz (pk-pk) | 97 ps @ 0.574 GHz (pk-pk) | N/A | N/A |
| Normalized periodic jitter (jitter/period) | 0.0586 | 0.055 | N/A | N/A |
| Power consumption (mW) | 0.13 | 2.71 | 50 | 5.5 |
| Active area (mm$^2$) | 0.09 | 0.014 | 0.4 | 0.075 |
| Technology | 22 nm FD-SOI | 28-nm FD-SOI | 45 nm SOI CMOS | 22 nm FD-SOI |
| FOM | 74.01 | 91.76 | 111.85 | 114.53 |

## 5. Conclusions

An XOR logic-based, low-power DLL frequency multiplier is presented. The circuit achieves 8× multiplication in three 2× stages. The design eliminates the need for edge combining, effectively minimizing the power consumption to 130 μW at 0.8 V supply. The circuit utilizes a new simpler duty cycle correction loop offering multiplication of a wide frequency. The duty cycle correction loop also ensures minimal duty cycle distortion across all PVT corners. A timing jitter of 24 ps (pk-pk) is observed at 2.44 GHz output comparable to the state-of-the-art options. The proposed design is a suitable low-power frequency multiplier for battery-less wireless sensor nodes.

**Author Contributions:** Conceptualization, N. and J.D.; methodology, N. and J.D.; software, N.; validation, N.; formal analysis, N.; investigation, N.; resources, J.D.; data curation, N.; writing—original draft preparation, N.; writing—review and editing, N. and J.D.; visualization, N. and J.D.; supervision, J.D.; project administration, J.D.; funding acquisition, J.D. All authors have read and agreed to the published version of the manuscript.

**Funding:** This research received no external funding.

**Data Availability Statement:** Data are contained within the article.

**Conflicts of Interest:** The authors declare no conflict of interest.

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
