# Peer review of "Design of a Low-Power Delay-Locked Loop-Based 8× Frequency Multiplier in 22 nm FDSOI"

_jlpea, doi:10.3390/jlpea13040064_

Round 1

Reviewer 1 Report

Comments and Suggestions for Authors

The manuscript discusses a low-power frequency multiplier utilizing a delay-locked loop (DLL) in 22-nm FDSOI technology. Notably, the authors employed current-starved delay cells to enhance power efficiency, resulting in decreased power consumption and a smaller chip area in comparison to traditional phase-locked loop (PLL) circuits. The design occupies an active area of 0.09 mm² and consumes just 130 μW at a 0.8 V supply voltage. While the manuscript is well-organized and well-written, there is room for improvement with the following suggestions:

  • To enhance the study's findings, consider including simulation results for 2X multiplication as depicted in Figure 2 and comparing them with the results in Figure 5.
  • It's advisable to incorporate a section about FDSOI technology in the revised manuscript, elucidating its specifications and justifying its selection over traditional CMOS technology.
  • Prior to Figure 7, include a block diagram and refer to various sections in your layout. The current layout lacks clarity regarding sub-circuits.
  • To emphasize the enhancements in your results, consider adding simulation outcomes to illustrate a 3X multiplication factor using your proposed topology, thereby strengthening the comparison presented in Table 1.

Reviewer 2 Report

Comments and Suggestions for Authors

Dear Authors,

Overall the paper is well organised.

1) please consider revisiting english as some mistakes remain

2) Some points need to be clariifed:

- You need to clearly state what are simulation results from what are measurement.

- As properly stated in the text, the architecture will be sensitive to matchning issue. I think that monte carlo siulations results should be welcomed

- On the architecture level : using a high resistance divider leads to a high impedance node sensitive to nise coupling. Can you comment on that?

- what is the constraints on the DC extraction filter ? its cut-off frequency, order and so on? How its characteristics affects jitter?

- You got pretty good results concerning jitter. Nevertheless, is the comparaison fair as the DLL multiplier ratio is lower than of the other implemnattions you are comparing with?.  I mean : how did you measure the DLL perforrmance? if it is with a 300Mhz signal generator?  May be a normalized comparison with an extaroplation of a same multiplying ration will be better for comparison.

Comments on the Quality of English Language

a few mistakes remain

Round 2

Reviewer 2 Report

Comments and Suggestions for Authors

Thanks for your feedback.